# Compatibility-aware Single-cell Continual Annotation

## Abstract

As massive well-labeled single-cell RNA-seq (scRNA-seq) data are available sequentially, automatic cell type annotation systems would require the model to update to expand their internal cell type library continuously. However, the model could suffer from the catastrophic forgetting phenomenon, in which the model's performance on the old tasks degrades significantly after it learns a new task. To enable the smooth upgrading of the system, the model must possess the ability to maintain performance on old tasks (stability) and adapt itself to learn new tasks (plasticity). We call such an updating process continual compatible learning. To adapt to this task, we propose a simple yet effective method termed scROD based on sample replay and objective decomposition. Specifically, we first maintain a memory buffer to save some cells from the previous tasks and replay them to learn together with the next incoming tasks. Then we decompose two training objectives in continual compatible learning, i.e., distinguishing new cell types from old ones and distinguishing between new ones, to avoid forgetting the model to varying degrees. Lastly, we assign distinct weights for two objectives to obtain a better trade-off between model stability and plasticity than the coupled approach. Comprehensive experiments on various benchmarks show that scROD can outperform existing scRNA-seq annotation methods and learn many cell types continually over a long period.

## 1 Introduction

The rapid development of single-cell RNA sequencing (scRNA-seq) technologies allows us to study tissue heterogeneity at the cellular level (Patel et al., 2014). Cell type annotation is one fundamental step in analyzing scRNA-seq data since many downstream cellular and gene-level analyses, such as cell-cell interaction and gene network analysis, are often based on specific cell types (Satija et al., 2015). Initially, single-cell communities annotated cell types through unsupervised cell clustering and differential gene expression analysis, which was gradually replaced by supervised cell classification methods as the scale of sequencing data grew larger. This approach is particularly evident and powerful in the era of deep learning (Cao et al., 2020b). In particular, cell classification in a universal scenario is often accomplished by mapping each cell onto a vector space using a function ("model") implemented by a deep neural network. The outputs of such a function in response to a cell are often represented as its embedding and prediction, and the prediction is usually calculated by the transformation of similarity between the cell embedding and cell type embedding (widely called prototype). A good embedding is expected to cluster cells belonging to the same cell type in the embedding space.

As cells of a new cell type become available, their embedding vectors are used to spawn a new cluster in the feature space, possibly modifying its metric to avoid crowding, in the form of lifelong learning or continual learning (Parisi et al., 2019). As time goes by, the annotation tasks grow, and the number of learned cell types increases with newly trained models. However, to preserve the acquired knowledge of old models, one has to train the new models by re-processing all task-related datasets that we have seen to recreate the clusters. Otherwise, the models would suffer from a phenomenon called catastrophic forgetting, where the performance of the model on the old tasks degrades significantly after it learns a new task (De Lange et al., 2021). Therefore, we aim to design an automatic cell type annotation system that enables new models to be deployed without forgetting previous knowledge and having to retrain all the tasks/datasets before. We call such a

process continual compatible training, and the model possesses the ability to maintain performance on old tasks/datasets (stability) and adapt itself to learn new tasks/datasets (plasticity). Nevertheless, an excess of stability or plasticity can interfere with the other, and hence the model needs to make a trade-off between stability and plasticity.

For continual compatible learning of the single-cell annotation system, we need to learn two objectives for each new dataset or task, including distinguishing new cell types from old cell types (i.e., new/old cell type distinction) and distinguishing between different new cell types (i.e., new cell type distinction). But these two training objectives may cause different degrees of forgetting in continual compatible learning and thus different trade-off strategies between model stability and plasticity are required for these two learning objectives. More specifically, if a new learning objective leads to more forgetting, a good continual compatible learner should pay more attention to the model's stability for this objective. On the contrary, if a new learning objective leads to less forgetting, a good continual compatible learner should pay more attention to the model's plasticity for this objective. However, when the annotation model mixes these different learning objectives, adjusting one of the learning objectives may influence others, inhibiting the model from achieving a good trade-off between stability and plasticity.

To address these issues, we propose a novel continual compatible annotation framework called scROD from the perspective of sample replay and objective decomposition. First, to avoid the overwriting of old cell types' knowledge in previous tasks by novel information from new tasks, we maintain a memory buffer to save some samples from the previous tasks and then use them to learn together with current samples. The exemplar method is a sample selection technique based on the nearest prototype classification confidence. It is worth noting that our exemplar set approximates the cell type prototype well and makes it possible to reduce redundant samples during the model's runtime. Second, by deeply analyzing the impacts of new/old cell type distinction and new cell type distinction, we find that these two learning objectives cause different degrees of forgetting. This evidence directly validates that mixing them is detrimental for the model to make a good trade-off between stability and plasticity. Third, we separate the two objectives for the new task by decomposing the loss of the new dataset. As a result, scROD can assign different weights for different objectives, which provides a way to obtain a better trade-off between stability and plasticity than the approach with coupled loss. To evaluate the performance of scROD fairly, we select massive large-scale scRNA-seq datasets and design comprehensive continual compatible annotation benchmarks. Extensive experiments on these benchmarks show that scROD settles the catastrophic forgetting problem effectively and can learn many cell types continually over a long period.

## 2 RELATED WORK

### 2.1 CELL TYPE ANNOTATION FOR SCRNA-SEQ DATA

Without losing generality, cell type annotations are mainly divided into manual annotation methods and automatic annotation methods (Pasquini et al., 2021). The former classifies cells by analyzing the differentially expressed genes of clusters to obtain marker genes with biological functions (Zhang et al., 2019), while the latter classifies cells by using supervised classification methods based on gene expression profiles (Alquicira-Hernandez et al., 2019). Considering the heavy workload of manual annotation methods in large-scale data, this paper focuses on continuous compatible learning of automated annotation methods. Recently, the single-cell community has seen a large number of automated annotation methods based on deep learning techniques (Flores et al., 2022). For example, scNym is a cell type classification model that uses semi-supervised and adversarial representation learning strategies (Kimmel & Kelley, 2021). scArches uses transfer learning to enable efficient, iterative reference building and contextualization of new datasets (Lotfollahi et al., 2022). SCALEX projects cells into a batch-invariant embedding space in a truly online manner without retraining the model (Xiong et al., 2022). CIForm is a Transformer-based cell-type annotation framework for scRNA-seq data that introduces the patch concept (Xu et al., 2023). scDOT combines distance metric learning and optimal transport to create a cell type annotation framework (Xiong & Zhang, 2024). However, none of these methods are actually suitable for our tasks, and we would demonstrate that they suffer from severe catastrophic forgetting problems.

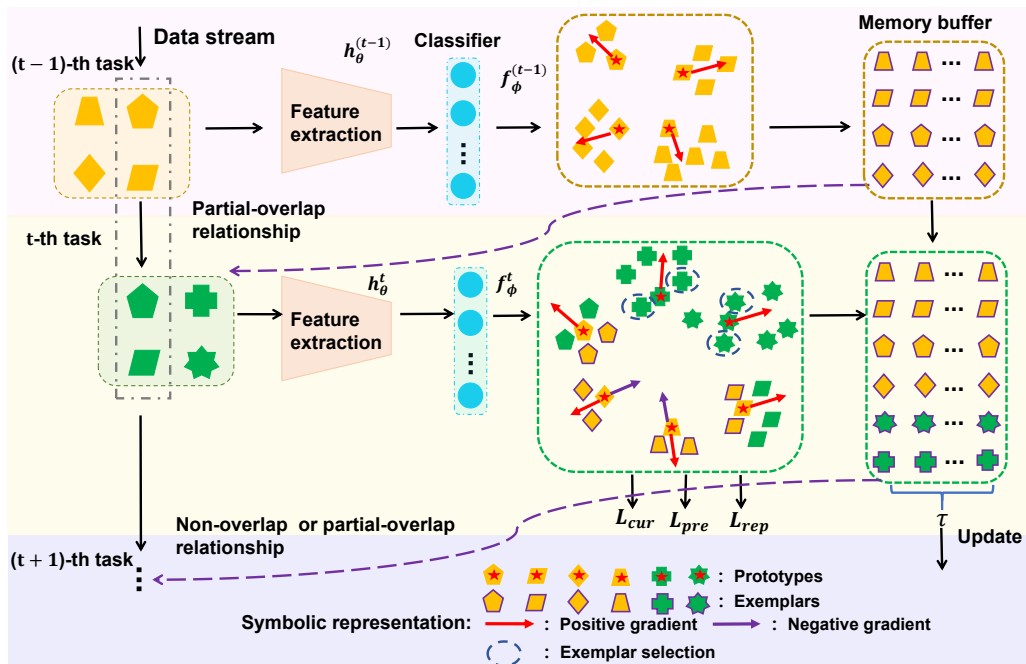

Figure 1: Schematics of scROD. The input of the model in the $t$-th task includes old samples stored in the memory buffer and the new data. Three loss function $\mathcal{L}_{cur}$, $\mathcal{L}_{pre}$ and $\mathcal{L}_{rep}$ are all based on the classifier's results. Specifically, $\mathcal{L}_{rep}$ replays the old knowledge by using exemplar sets. $\mathcal{L}_{pre}$ is related to new/old cell type distinction and $\mathcal{L}_{cur}$ is related to new cell type distinction.

## 2.2 CONTINUAL LEARNING AND COMPATIBLE LEARNING

Continual learning deals with cases where an existing model evolves over time (Masana et al., 2022). In (Li & Hoiem, 2017), model distillation is used as a form of regularization when introducing new classes. In (Rebuffi et al., 2017), old class centers are used to regularize samples from the new classes. Methods addressing catastrophic forgetting are most closely related to our work, as a common reason for forgetting is the changing of the embedding feature for the subsequent classifiers. The concept "compatibility" is a design characteristic considered in software engineering (Nagarakatte et al., 2009). Forward compatibility allows a system to accept input intended for a later version of itself (Zhou et al., 2022), and backward compatibility allows for interoperability with an older legacy system (Srivastava et al., 2020). BCT is an algorithm that allows new embedding models to be compatible with old models (Shen et al., 2020). Other works attempt to construct a unified representation space on which models are compatible (Hu et al., 2022). These procedures also modify the training of individual models to ensure that they are easy to transform into this unified embedding space. In this paper, our task incrementally trains the new model and allows the old sample to be compatible with the new sample in the feature space of the new model.

## 3 METHOD

### 3.1 PROBLEM FORMULATION

We begin with problem setting and notations. In continual compatible annotation scenarios, scRNA-seq data are seen in a data stream and are learned by the model in sequential order, i.e., sample sets $\{\mathcal{X}_1, \mathcal{X}_2, ...\}$ with label sets $\{\mathcal{Y}_1, \mathcal{Y}_2, ...\}$. They can come from the same or different scRNA-seq datasets. We use $\mathcal{D}_t = \{x_i^t, y_i^t\}_{i=1}^{N_t}$ to denote the training dataset of the $t$-th task, where $N_t$ is the number of cells for task $t$. For convenience, we assume that the specific cell type set of $t$-th task is $\mathcal{C}_t$. The label space relationship among datasets seen in the different tasks can be non-overlap or partial-overlap.

Our model consists of a feature extractor $h_\theta$ with the parameter set $\theta$ and a classifier $f_\phi$ with the parameter set $\phi$ (see Figure 1). Given a cell $x$, the model produces the annotation logits $o(x; \theta, \phi) = f_\phi(h_\theta(x))$, which is used to calculate the training loss or to predict the cell type label in testing. Now we introduce the gradient-based analysis on logits in continual comparible learning. Specifically, for the $t$-th task, the model is usually learned by minimizing the softmax-based cross-entropy loss,

$$\mathcal{L}_{ce} = -\sum_{i \in \mathcal{D}_t} \log(p_{y_i}), \quad p_{y_i} = \frac{\exp(o_{y_i})}{\sum_{j=1}^{|\cup_{l=1}^t \mathcal{C}_l|} \exp(o_j)}, \tag{1}$$

where $|\cup_{l=1}^t \mathcal{C}_l|$ is the number of cell types that the model has seen until $t$-th task. Given a training sample $x$ of cell type $y_i$, the gradients on logits ($y_j \neq y_i$) are given by,

$$\frac{\partial \mathcal{L}_{ce}(o(x; \theta, \phi))}{\partial o_{y_i}} = p_{y_i} - 1, \quad \frac{\partial \mathcal{L}_{ce}(o(x; \theta, \phi))}{\partial o_{y_j}} = p_{y_j}. \tag{2}$$

From the above equation, we can see that $x$ gives its true logit $o_{y_i}$ a negative gradient and other logits $o_{y_j}$ positive gradient. As the gradient update rule for a parameter $w$ is $w = w - lr * \nabla w$, where $lr$ is the learning rate. The negative gradient $p_{y_i} - 1$ results in an increase in $o_{y_i}$ for the true cell type $y_i$ and the positive gradient $p_{y_j}$ results in a decrease in $o_{y_j}$ for each wrong cell type $y_j$. Thus, the negative gradient encourages the model to output a larger probability for the true cell type and positive gradients help output lower probabilities for the wrong cell types. However, as the model has no access to the training data of previous tasks when it learns a new task continually, all gradients on previous cell types are positive during the new task training, i.e., an imbalance of positive and negative gradients. Then the model tends to output smaller probabilities on the previous cell types, biasing the classification towards the new cell types.

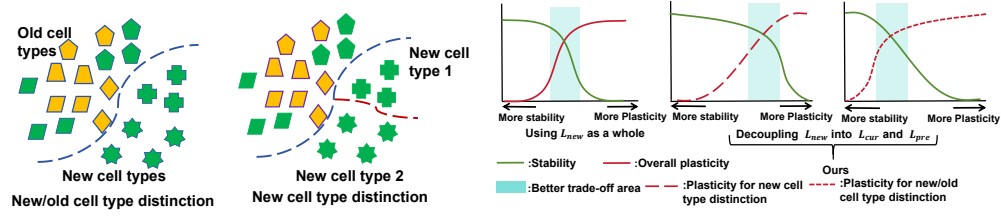

(a) Different learning objectives      (b) Stability-plasticity trade-off in different strategies

Figure 2: (a) Two different learning objects with loss funtion $\mathcal{L}_{pre}$ and $\mathcal{L}_{cur}$. (b) Contrast diagram of using $\mathcal{L}_{new}$ as a whole and decomposition of $\mathcal{L}_{new}$ into $\mathcal{L}_{cur}$ and $\mathcal{L}_{pre}$. y-axis represents the model's abilities, including plasticity and stability.

## 3.2 Constructing Memory Buffer

Based on the above analysis, we can argue that storing some samples that have appeared before is a necessary step in balancing gradient propagation and thus preventing the catastrophic forgetting problem. However, two conditions for storing samples should be considered. On the one hand, the manner of storing all the train samples that have appeared will lead to large memory requirements as the number of samples previously learned increases. On the other hand, if samples are stored randomly without regard to their cell types, some cell types may have no samples saved, causing models to perform poorly on them. Taking these two conditions into account, we need to select a subset of samples in the current task carefully as exemplar samples saved in a memory buffer $\mathcal{M}$ after finishing every task, and then replay the whole memory buffer $\mathcal{M}$ to join the next stage training. Taking the $t$-th task as an example, the set of cell types that have been observed before can be denoted as $\{\mathcal{C}_1, \mathcal{C}_2, ..., \mathcal{C}_{t-1}\}$, respectively, and the number of cell types newly added at the $t$-th stage can be denoted as $\kappa_t = |\mathcal{C}_t \setminus (\cup_{k=1}^{t-1} \mathcal{C}_k)|$. Then, the exemplar sample sets at the $t$-th stage $\{E_1^t, E_2^t, ..., E_{\kappa_t}^t\}$ should be constructed dynamically out of the data stream. After the training of $t$-th task, we can add these exemplar sample sets into the memory buffer $\mathcal{M}$ and update it. To control the size of memory requirements, we assume that the number of exemplar samples for every cell type is fixed as a hyperparameter $\tau$. In the process of exemplar selection, it is assumed that the selected exemplars should be sufficiently close to the corresponding cell type center, thereby creating a representative set of samples from such a distribution. Specifically, for any cell type $y_i$ at the $t$-stage, we can obtain the

logit of each sample $x_i$ that belongs to this cell type, which can be expressed as $o(x_i)_{y_i}$. Then, we can select the top $\tau$ samples with the largest logit as exemplars for this cell type. Moreover, for those old cell types that were learned before, the exemplar set is not reselected at the current stage.

### 3.3 ANALYZING LEARNING OBJECTIVES

After maintaining a memory buffer $\mathcal{M}$ with a limited size to store a small portion of old samples, we can combine them with the new data in the next task to update and upgrade the model. Specifically, when receiving a mini-batch of new cells $\mathcal{B}_t$ from a new task $t$, the model retrieves a mini-batch of samples $\mathcal{B}_\mathcal{M}$ from $\mathcal{M}$ and replays them with the new samples $\mathcal{B}_t$ to achieve a trade-off between stability and plasticity. The losses used in our model can be written as follows,

$$\mathcal{L}_{cls} = \frac{1}{|\mathcal{B}_t|} \sum_{i=1}^{|\mathcal{B}_t|} \mathcal{L}_{new}(f_\phi(h_\theta(x_i^t)), y_i^t) + \frac{1}{|\mathcal{B}_\mathcal{M}|} \sum_{i=1}^{|\mathcal{B}_\mathcal{M}|} \mathcal{L}_{rep}(f_\phi(h_\theta(x_i^\mathcal{M})), y_i^\mathcal{M}). \tag{3}$$

Here, $\mathcal{L}_{new}$ is the loss for the new task and is mainly for the plasticity of the model. We can use the cross-entropy loss like Equation 1 to define it. In contrast, $\mathcal{L}_{rep}$ is the replay loss and is mainly for the stability of the model. For it, we can use the cross-entropy loss that is only constrained to the previous cell types before task $t$, i.e.,

$$\mathcal{L}_{rep}(f_\phi(h_\theta(x_i^\mathcal{M})), y_i^\mathcal{M}) = -\log(\frac{\exp(o_{y_i^\mathcal{M}})}{\sum_{j=1}^{|\cup_{l=1}^{t-1} \mathcal{C}_l|} \exp(o_j)}). \tag{4}$$

But one detail to note is that $\mathcal{L}_{new}$ is not only related to new/old cell type distinction but also related to new cell type distinction. Actually, they are two different learning objectives. Therefore, we decompose the loss $\mathcal{L}_{new}$ according to the two learning objectives as follows,

$$\mathcal{L}_{new}(f_\phi(h_\theta(x)), y) = -\log(\frac{\exp(o_{y_i})}{\sum_{j=1}^{|\cup_{l=1}^{t} \mathcal{C}_l|} \exp(o_j)}) \tag{5}$$

$$= -\log(\frac{\exp(o_{y_i})}{\sum_{j=1}^{|\mathcal{C}_t \setminus \cup_{l=1}^{t-1} \mathcal{C}_l|} \exp(o_j)}) - \log(\frac{\sum_{j=1}^{|\mathcal{C}_t \setminus \cup_{l=1}^{t-1} \mathcal{C}_l|} \exp(o_j)}{\sum_{j=1}^{|\cup_{l=1}^{t} \mathcal{C}_l|} \exp(o_j)}) \tag{6}$$

$$= \mathcal{L}_{cur}(f_\phi(h_\theta(x)), y; \mathcal{C}_t \setminus \cup_{l=1}^{t-1} \mathcal{C}_l) + \mathcal{L}_{pre}(f_\phi(h_\theta(x))). \tag{7}$$

Here, we use $\mathcal{L}_{cur}(\cdot; \mathcal{C}_t \setminus \cup_{l=1}^{t-1} \mathcal{C}_l)$ to denote the cross-entropy loss restricted to new cell types. Obviously, $\mathcal{L}_{cur}(f_\phi(h_\theta(x)), y; \mathcal{C}_t \setminus \cup_{l=1}^{t-1} \mathcal{C}_l)$ is related to new cell type distinction; $\mathcal{L}_{pre}(f_\phi(h_\theta(x)))$ is related to new/old cell type distinction (see Figure 2(a)). Note that both $\mathcal{L}_{cur}(f_\phi(h_\theta(x)), y; \mathcal{C}_t \setminus \cup_{l=1}^{t-1} \mathcal{C}_l)$ and $\mathcal{L}_{pre}(f_\phi(h_\theta(x)))$ are for the plasticity of the model and may cause catastrophic forgetting. Furthermore, these two losses have the same weight in Equation 3 due to the coupling property.

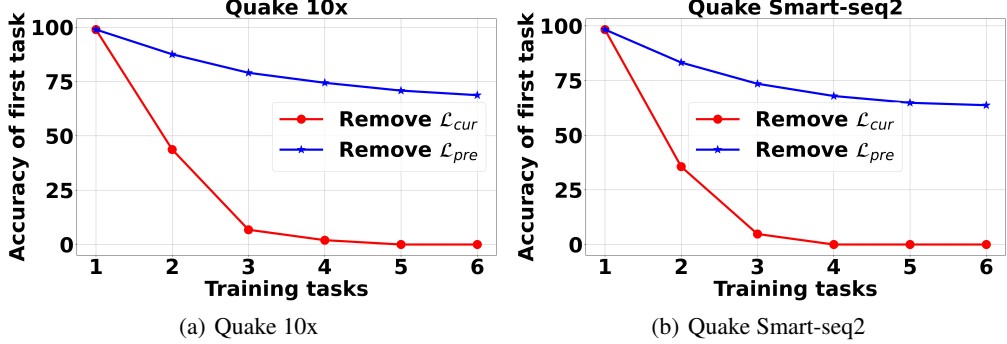

(a) Quake 10x            (b) Quake Smart-seq2

Figure 3: (a & b ) The variation of the first task's accuracy with the training task number on different datasets.

To evaluate the impact of $\mathcal{L}_{cur}(\cdot; \mathcal{C}_t \setminus \cup_{l=1}^{t-1} \mathcal{C}_l)$ and $\mathcal{L}_{pre}(\cdot)$, we conduct control experiments on two datasets Quake 10x and Quake Smart-seq2 from Tabula Muris atlas. Specifically, we first let the model learn on the first task through valina cross-entropy loss. Then, before learning the subsequent tasks, we remove one of the two losses in Equation 7 and analyze the forgetting of the first task.

Table 1: Comparative analysis of performance among diverse baselines in intra-tissue continual compatible annotation benchmark.

| | Task 1 (Baron_human) | | | Task 2 (Enge) | | | Task 3 (Muraro) | | | Task 4 (Segerstolpe) | | |
|---|---|---|---|---|---|---|---|---|---|---|---|---|
| Pancreas tissue | old | new | overall | old | new | overall | old | new | overall | old | new | overall |
| **Finetune** | - | 98.0 | 98.0 | 86.1 | 94.7 | 87.9 | 93.3 | **96.8** | 93.9 | 83.1 | 92.9 | 83.8 |
| **Joint** | - | 98.0 | 98.0 | 97.7 | **96.5** | 97.5 | **97.5** | 95.8 | **97.2** | 97.2 | 98.9 | **97.4** |
| **scNym** | - | 97.4 | 97.4 | 94.5 | 95.3 | 94.6 | 95.7 | 95.7 | 95.7 | 92.8 | 95.9 | 93.1 |
| **scArches** | - | 96.1 | 96.1 | 61.3 | 78.7 | 65.0 | 78.3 | 79.2 | 78.4 | 77.9 | 73.6 | 77.6 |
| **SCALEX** | - | 92.9 | 92.9 | 76.2 | 72.5 | 75.4 | 71.3 | 65.8 | 70.4 | 66.0 | 64.5 | 65.9 |
| **CIForm** | - | 98.4 | 98.4 | 79.6 | 94.2 | 82.7 | 92.6 | 96.6 | 93.3 | 82.8 | 92.5 | 83.5 |
| **scDOT** | - | 94.5 | 94.5 | 64.7 | 93.1 | 71.9 | 80.5 | 83.8 | 81.3 | 80.1 | 85.6 | 81.5 |
| **Replay** | - | 98.0 | 98.0 | 97.9 | 96.4 | **97.6** | 97.1 | 96.0 | 97.0 | 96.8 | **99.2** | 97.0 |
| **scROD** | - | **98.2** | **98.2** | **98.0** | 96.2 | **97.6** | 97.1 | 96.2 | 97.0 | 96.8 | **99.2** | 97.0 |

Table 2: Comparative analysis of performance among diverse baselines in inter-tissue continual compatible annotation benchmark.

| | Task 1 (Eye) | | | Task 2 (Intestine) | | | Task 3 (Pancreas) | | | Task 4 (Stomach) | | |
|---|---|---|---|---|---|---|---|---|---|---|---|---|
| Cao atlas | old | new | overall | old | new | overall | old | new | overall | old | new | overall |
| **Finetune** | - | 98.6 | 98.6 | 3.4 | 97.8 | 59.0 | 11.7 | 97.6 | 41.0 | 30.2 | **96.5** | 34.2 |
| **Joint** | - | 97.5 | 97.5 | **97.8** | 96.2 | 96.9 | **96.4** | 94.9 | **95.9** | 95.6 | 84.4 | **94.9** |
| **scNym** | - | **99.5** | **99.5** | 71.1 | **98.4** | 87.2 | 81.4 | 93.9 | 85.7 | 75.2 | 86.8 | 75.9 |
| **scArches** | - | 99.4 | 99.4 | 68.0 | 97.4 | 85.3 | 82.9 | 93.4 | 86.5 | 76.1 | 77.2 | 76.2 |
| **SCALEX** | - | 97.9 | 97.9 | 32.4 | 79.4 | 60.1 | 60.7 | 74.5 | 65.4 | 58.2 | 34.4 | 56.8 |
| **CIForm** | - | 97.4 | 97.4 | 53.6 | 97.2 | 80.1 | 78.4 | 97.4 | 84.2 | 76.7 | 93.2 | 80.8 |
| **scDOT** | - | 98.5 | 98.5 | 47.3 | 96.2 | 76.6 | 70.1 | 95.3 | 74.9 | 71.2 | 91.5 | 74.5 |
| **Replay** | - | 98.6 | 98.6 | 93.3 | 97.9 | 96.0 | 80.7 | **97.7** | 86.5 | 87.3 | 94.4 | 87.7 |
| **scROD** | - | 99.1 | 99.1 | 97.7 | 97.4 | **97.5** | 88.2 | 97.0 | 91.2 | 89.3 | 92.6 | 89.5 |

The experimental results in Figure 3 show the annotation accuracy of the first task when the model learns subsequent tasks. We can see that removing $\mathcal{L}_{pre}(\cdot)$ results in less forgetting of the first task than removing $\mathcal{L}_{cur}(\cdot; \mathcal{C}_t \setminus \cup_{l=1}^{t-1}\mathcal{C}_l)$. In other words, $\mathcal{L}_{pre}(\cdot)$ leads to more forgetting than $\mathcal{L}_{cur}(\cdot; \mathcal{C}_t \setminus \cup_{l=1}^{t-1}\mathcal{C}_l)$. It is intuitively reasonable for these results. Since our method keeps limited samples in the memory buffer when the model learns a new task, it has access to much fewer samples from the old cell types than from the new cell types. So utilizing loss $\mathcal{L}_{pre}(\cdot)$ to learn to distinguish between new cell types and old cell types introduces a risk of biasing the model towards the new cell types, potentially leading to serious catastrophic forgetting. In contrast, loss $\mathcal{L}_{cur}(\cdot; \mathcal{C}_t \setminus \cup_{l=1}^{t-1}\mathcal{C}_l)$ is independent of the old cell types, thereby avoiding introducing a risk of biasing the model towards the new cell types. In particular, based on this analysis, we can conclude that a good continual compatible learner should assign a larger weight to $\mathcal{L}_{cur}(\cdot; \mathcal{C}_t \setminus \cup_{l=1}^{t-1}\mathcal{C}_l)$ and a smaller weight to $\mathcal{L}_{pre}(\cdot)$. However, loss in Equation 3 fails to achieve this goal due to the coupling property.

### 3.4 DECOMPOSING OBJECTIVE FUNCTION

The last section has demonstrated the impact of different learning objectives on the model's forgetting and the issue of the coupling property. To address this issue, we propose a new strategy called objective decomposition to remove the coupling property. Specifically, our method uses the following loss to perform continual compatible learning,

$$\mathcal{L}_{cls} = \frac{1}{|\mathcal{B}_t|} \sum_{i=1}^{|\mathcal{B}_t|} (\alpha_1 \mathcal{L}_{cur}(f_\phi(h_\theta(x)), y; \mathcal{C}_t \setminus \cup_{l=1}^{t-1}\mathcal{C}_l) \tag{8}$$

$$+ \alpha_2 \mathcal{L}_{pre}(f_\phi(h_\theta(x)))) + \frac{1}{|\mathcal{B}_\mathcal{M}|} \sum_{i=1}^{|\mathcal{B}_\mathcal{M}|} \mathcal{L}_{rep}(f_\phi(h_\theta(x_i^\mathcal{M})), y_i^\mathcal{M}),$$

where $\alpha_1$ and $\alpha_2$ are two coefficients that control the weight of the two different learning objectives (see Figure 2(b)). The finding in the last section tells us that we should set $\alpha_2$ to be smaller than $\alpha_1$, to make the model achieve a better trade-off between stability and plasticity than the approach with

Table 3: Comparative analysis of performance among diverse baselines in inter-data continual compatible annotation benchmark.

| Mixed atlas | Task 1 (He) | | | Task 2 (Madissoon) | | | Task 3 (Stewart) | | | Task 4 (Vento) | | |
|---|---|---|---|---|---|---|---|---|---|---|---|---|
| | old | new | overall | old | new | overall | old | new | overall | old | new | overall |
| **Finetune** | - | 78.7 | 78.7 | 2.0 | 90.9 | 72.7 | 37.1 | 95.1 | 51.2 | 22.6 | **97.9** | 52.7 |
| **Joint** | - | 78.7 | 78.7 | 79.0 | 90.9 | 88.5 | **88.5** | 93.3 | **89.7** | **88.6** | 96.1 | **91.6** |
| **scNym** | - | **83.2** | **83.2** | 34.0 | 90.4 | 78.9 | 75.6 | 87.4 | 78.5 | 69.2 | 91.2 | 78.0 |
| **scArches** | - | 74.9 | 74.9 | 35.3 | 88.6 | 77.7 | 67.7 | 85.2 | 71.9 | 70.5 | 90.8 | 78.6 |
| **SCALEX** | - | 78.2 | 78.2 | 10.0 | 85.4 | 70.0 | 60.8 | 60.7 | 60.8 | 63.2 | 81.3 | 70.4 |
| **CIForm** | - | 81.5 | 81.5 | 48.7 | 90.2 | 80.6 | 73.1 | 93.9 | 79.7 | 71.4 | 94.0 | 79.2 |
| **scDOT** | - | 77.9 | 77.9 | 39.0 | 89.2 | 78.1 | 74.5 | 89.6 | 78.8 | 71.3 | 92.4 | 78.3 |
| **Replay** | - | 79.2 | 79.2 | 78.9 | 90.9 | 88.5 | 83.4 | **95.4** | 86.3 | 72.1 | 97.6 | 82.3 |
| **scROD** | - | 80.2 | 80.2 | **81.0** | **91.1** | **89.0** | 86.3 | 95.0 | 88.5 | 78.8 | 97.0 | 86.1 |

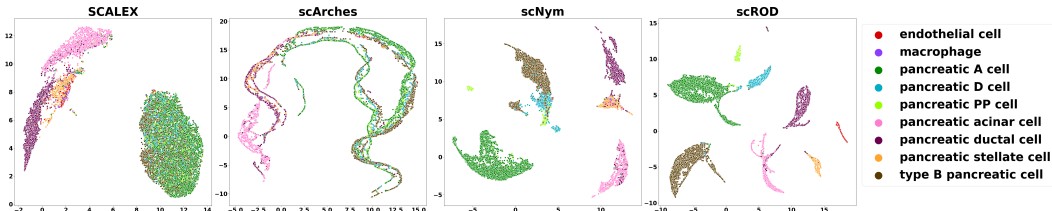

Figure 4: UMAP visualization of test data for four methods on the intra-tissue benchmark after the fourth task.

coupled loss. Furthermore, since $\mathcal{L}_{pre}(\cdot)$ is for new/old cell type distinction, we set $\alpha_2$ proportional to the ratio $\frac{|\mathcal{C}_t \setminus \cup_{l=1}^{t-1} \mathcal{C}_l|}{|\cup_{l=1}^{t-1} \mathcal{C}_l|}$ to make the model not bias toward old or new cell types, i.e., $\alpha_2 = \rho \frac{|\mathcal{C}_t \setminus \cup_{l=1}^{t-1} \mathcal{C}_l|}{|\cup_{l=1}^{t-1} \mathcal{C}_l|}$, where $\rho$ is a hyperparameter. In contrast, since $\mathcal{L}_{cur}(\cdot; \mathcal{C}_t \setminus \cup_{l=1}^{t-1} \mathcal{C}_l)$ is only related to the new cell types, we set $\alpha_1$ to be a constant value. Note that when the number of tasks increases, the number of old cell types also increases. In particular, when the number of old tasks is large, the number of old cell types $|\cup_{l=1}^{t-1} \mathcal{C}_l|$ is usually much larger than the number of new cell types $\mathcal{C}_t \setminus \cup_{l=1}^{t-1} \mathcal{C}_l$. At this time, $\alpha_2$ is much smaller than $\alpha_1$. Setting $\alpha_2$ to be as large as $\alpha_1$, or setting $\alpha_1$ to be as small as $\alpha_2$ fails to make the model achieve a good trade-off between stability and plasticity, which will be verified in the experiments.

## 4 EXPERIMENTS

### 4.1 EXPERIMENTAL SETTINGS

**Datasets and Metrics.** To simulate the continual compatible learning of scRNA-seq annotation systems, we design three types of annotation scenarios: intra-tissue annotation, inter-tissue annotation, and inter-data annotation. For the first one, we select four datasets from pancreatic tissue generated by different sequencing technologies, namely Baron_human (Baron et al., 2016), Enge (Enge et al., 2017), Muraro (Muraro et al., 2016), and Segerstolpe (Segerstolpe et al., 2016). They share most of the cell types, especially for Baron_human, which includes almost all cell types in the other three datasets. For the second one, we use a large-scale atlas dataset called Cao (Cao et al., 2020a) and select four tissues from it, i.e., Eye, Intestine, Pancreas, and Stomach. Compared with the intra-tissue setting, only a small number of cell types are shared between the four tissues, which can easily lead to catastrophic forgetting. For the third one, we choose four large-scale datasets that are sequenced by various tissues and technologies, namely He (He et al., 2021), Madissoon (Madissoon et al., 2020), Stewart (Stewart et al., 2019), and Vento (Vento-Tormo et al., 2018). It is worth noting that there is a strong batch effect between them, which directly affects the accuracy of annotations. In each experimental setting, we learn the cell type knowledge from each dataset sequentially, i.e., a total of four stages. Unless otherwise noted, the train set and test set are split according to the ratio 1:9 in each stage, i.e., labeled ratio=0.1. At each stage, we calculate three types of accuracy: the annotation accuracy on the test set in all previous stages, that is, the old accuracy, which quantitatively expresses

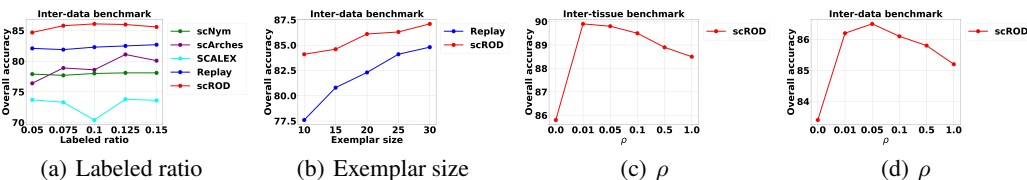

Figure 5: Overall accuracy on inter-tissue and inter-data benchmarks. (a) Varying the labeled ratio; (b) Varying the exemplar size; (c, d) Varying the parameter $\rho$.

the stability of the model; the annotation accuracy on the test set in the current stage, that is, the new accuracy, which quantitatively expresses the plasticity of the model; the annotation accuracy on all test sets up to the current stage, that is, the overall accuracy, which quantitatively expresses the trade-off between stability and plasticity. The accuracy in the result tables is the average of three runs.

**Comparison baselines.** We first select four state-of-the-art deep single-cell annotation methods, scNym (Kimmel & Kelley, 2021), scArches (Lotfollahi et al., 2022), CIForm (Xu et al., 2023), and scDOT (Xiong & Zhang, 2024), for comparison to illustrate that they are not directly adapted to our tasks. At the same time, we also compare with SCALEX (Xiong et al., 2022), an online annotation method, which claims to be able to project cells into a common embedding space without retraining the model. Without loss of generality, the results obtained by the three methods are all run under their default parameter settings. To confirm the advantage of our objective decomposition strategy, we also use the standard cross-entropy function as the training objective, and this baseline is denoted as Replay. We also include two methods without continual learning, Joint and Finetune, in the comparison. Here, Joint denotes the method that learns all the tasks jointly while Finetune denotes the method that learns all the tasks sequentially without any sample replay. The accuracy of Joint can be treated as the accuracy upper-bound and the accuracy of Finetune can be treated as the accuracy lower-bound.

**Implementation details.** Our algorithm is implemented by PyTorch and we conduct all experiments on one Tesla A100 GPU. Similar to scNym, scArches, and SCALEX, we also use the denoising autoencoder as our basic network (Eraslan et al., 2019). The encoder consists of two fully connected layers with sizes 512 and 256 respectively. The size of the low-dimensional latent space is 128, on top of which we externally attach a prototype-based classifier. The decoder is a symmetrical structure to the encoder and also consists of two fully connected layers with sizes of 256 and 512 respectively. The training batch size is set to 256 and the optimizer is Adam with a learning rate of 1e-4. The exemplar size $\tau$ for each learned cell type is set to 20 by default. We use the weight hyperparameters $\alpha_1 = 1.0$ and $\rho = 0.1$ in classification loss for model training. For each continual learning stage, the whole model is updated for 200 epochs. Subsequent stages utilize the checkpoint from the terminating stage to initiate the model.

### 4.2 EXPERIMENTAL RESULTS

**Intra-tissue benchmark.** Table 1 shows the accuracy of each method in four sequential tasks on pancreatic tissue. By comparing the old and overall accuracy of Finetune and Joint, it is not difficult to see that even when fewer new cell types appear in each stage, continual compatible annotation will still have a slight catastrophic forgetting problem. However, after we use the memory buffer to store a very small number of samples, the replay strategy immediately alleviates this forgetting issue, and scROD also performs well. Both of them find a good trade-off between remembering old knowledge and accumulating new knowledge. Among the other three annotation methods, we found that scNym performed relatively competitively, followed by CIForm, scDOT, and scArches. On the contrary, SCALEX, which was customized for online annotation, performed less satisfactorily. The main reason for this phenomenon is that scNym learns a low-dimensional latent space that is more suitable for continual annotation, while the embedding representation learned by SCALEX in the absence of training data from the previous stage cannot accurately separate old cell types and new cell types.

**Inter-tissue benchmark.** We turn to observations of continued compatible annotation scenarios across tissues where new cell types emerge frequently. Table 2 shows the annotation accuracy of

each method in four different tissues on the Cao atlas. First of all, compared with the results in the intra-tissue benchmark, Finetune's accuracy on the old task dropped off a cliff starting from the second task. Although Finetune's new accuracy is excellent compared to Joint, its overall accuracy rate is almost unacceptable. However, Replay alleviates the collapse of the old accuracy to a certain extent by replaying a small number of old samples, while scROD further improves the old accuracy based on it by decomposing the loss function, achieving a better balance between stability and plasticity. Interestingly, by comparing Joint and scROD, we find that the new accuracy of scROD is higher, and the old accuracy of Joint is higher, which shows that massive samples of old cell types will also restrict the annotation of new cell types. Similar to the results of the intra-tissue setting, scNym performs better than the other four tested annotation algorithms but is still far inferior to our method. This evidence suggests that existing annotation methods are not well suited to the task of continual compatible annotation across tissues.

**Inter-data benchmark.** Next, we analyze the challenging task of continual compatible annotation in the inter-data setting, where there are serious batch effects between datasets. Judging from the results in Table 3, the overall accuracy of almost all methods has declined compared with the former two benchmarks, which shows that the batch effect affects the accuracy of the annotation process. However, compared with Finetune and other annotation algorithms, our methods Replay and scROD can still solve the catastrophic forgetting problem well, especially scROD, which not only benefits from the sample replay strategy but also benefits from decoupling the learning objectives of old and new tasks. In addition, we can also find that scROD even performs better than the Joint baseline on some tasks. This is not surprising because, for such a large-scale continuous compatibility task, a subset of cells at classification boundaries can confuse the discriminative ability of the model. Although scNym, CIForm, scDOT, and scArches still outperform SCALEX on this benchmark, it is obvious that the gap between them is relatively smaller than the gap on the former two benchmarks. This may also be due to the fact that these methods have difficulty learning discriminative feature representations on large-scale benchmarks that carry severe batch effects.

**Feature visualization.** To further observe the annotation result of each method after continual learning intuitively, we extract their low-dimensional embedding features and use the UMAP approach to visualize them in the two-dimensional plane. Figure 4 shows the UMAP plots of four methods' test data on the intra-tissue benchmark after training the last task. We can see that SCALEX and scArches mix most different cell types, seriously compromising the plasticity and stability of the model. scNym performs better than them but still does not separate pancreatic D cells, pancreatic PP cells, and type B pancreatic cells clearly. On the contrary, scROD performs well, effectively distances different cell types, and avoids forgetting problems when the model continuously learns multiple tasks. In addition, we also present the visualization of scROD after each task in the supplementary.

**Robustness analysis.** We first discuss the effect of the labeled ratio on the model, which controls the ratio of train and test data in each task. We set its value in the range of $[0.05, 0.075, 0.1, 0.125, 0.15]$. Figure 5(a) shows the trends of the overall accuracy at the fourth tasks of scROD and other baselines on inter-data benchmarks, respectively. It can be seen that the overall accuracy of scROD is relatively stable with respect to the labeled ratio, indicating that the effect of the labeled ratio on our method is slight. Moreover, scROD maintains satisfactory performance among the compared methods, validating its superiority in preventing catastrophic forgetting and resisting the batch effect.

Then we study the impact of $\tau$ that controls the number of exemplars stored for each cell type. We also conduct experiments on inter-data benchmarks and give the variation of the overall accuracy for Replay and scROD at the fourth task in Figure 5(b). The value of $\tau$ ranges from 10 to 30 and the results show that the overall accuracy increases as $\tau$ increases for both methods, indicating that we need to balance the model precision and computational burden in practicality. We can also see that when the value of $\tau$ is small, such as 10, scROD can still provide excellent performance, validating the superiority of scROD under an extremely limited memory buffer.

**Ablation study.** We change the value of $\alpha_1$ and $\alpha_2$ to show the effectiveness of setting $\alpha_1 = 1$ and $\alpha_2 = 0.1 \frac{|\mathcal{C}_t \setminus \cup_{l=1}^{t-1} \mathcal{C}_l|}{|\cup_{l=1}^{t-1} \mathcal{C}_l|}$. We first set the value of $\alpha_1 = \alpha_2$ to remove the decoupling property. There are two possibilities to set $\alpha_1 = \alpha_2$. The first possibility is to set $\alpha_1 = \alpha_2 = 1$ and the second possibility is to set $\alpha_1 = \alpha_2 = 0.1 \frac{|\mathcal{C}_t \setminus \cup_{l=1}^{t-1} \mathcal{C}_l|}{|\cup_{l=1}^{t-1} \mathcal{C}_l|}$. Table 4 shows the results of these two possibilities, which are significantly inferior to our method. This indicates that separating the two different objectives by decomposing the loss of the new task is necessary for the model to achieve

Table 4: Ablation study for $\alpha_1$ and $\alpha_2$ on the inter-tissue and inter-data benchmarks, where $\frac{\mathcal{C}_{new}}{\mathcal{C}_{old}} = \frac{|\mathcal{C}_t \setminus \cup_{l=1}^{t-1} \mathcal{C}_l|}{|\cup_{l=1}^{t-1} \mathcal{C}_l|}$.

| | inter-tissue | | | inter-data | | |
|---|---|---|---|---|---|---|
| Choice | old | new | overall | old | new | overall |
| $\alpha_1 = 1, \alpha_2 = 0.1 \frac{\mathcal{C}_{new}}{\mathcal{C}_{old}}$ | 89.3 | 92.6 | 89.5 | 78.8 | 97.0 | 86.1 |
| $\alpha_1 = 1, \alpha_2 = 1$ | 88.2 | 93.6 | 88.5 | 76.9 | 97.5 | 85.2 |
| $\alpha_1 = 0.1 \frac{\mathcal{C}_{new}}{\mathcal{C}_{old}}, \alpha_2 = 0.1 \frac{\mathcal{C}_{new}}{\mathcal{C}_{old}}$ | 87.7 | 91.8 | 87.3 | 76.3 | 96.6 | 84.4 |
| $\alpha_1 = 0.1 \frac{\mathcal{C}_{new}}{\mathcal{C}_{old}}, \alpha_2 = 1$ | 86.0 | 28.9 | 82.6 | 81.0 | 55.7 | 70.9 |

good performance. In Table 4, we also show the result of a variant by exchanging the value of $\alpha_1$ and $\alpha_2$, i.e., $\alpha = 0.1 \frac{|\mathcal{C}_t \setminus \cup_{l=1}^{t-1} \mathcal{C}_l|}{|\cup_{l=1}^{t-1} \mathcal{C}_l|}$ and $\alpha_2 = 1$. We can find that the performance of this variant is still significantly inferior to our method.

**Hyperparameter sensitivity.** We vary the value of $\rho$ in the $\alpha_2$ setting to show its impact on the performance of the model. Figure 5(c) and Figure 5(d) give the analysis of the inter-tissue and inter-data benchmarks. Note that when $\rho = 0$, $\alpha_2 = 0$ and the weight of $\mathcal{L}_{pre}$ is always zero. At this time, the $\mathcal{L}_{new}$ degenerates to the situation where the model focuses on new cell type distinction. When the value of $\rho$ increases, $\alpha_2$ also increases, and the performance of the model first increases and then decreases. This phenomenon is reasonable since a larger weight for $\mathcal{L}_{pre}$ leads to more forgetting and thus influences the overall model performance.

## 5 CONCLUSION

In this paper, we propose a novel method called scROD for continual compatible learning of scRNA-seq data. scROD introduces the concepts of sample replay and objective decomposition to alleviate the catastrophic forgetting problem encountered by annotation systems during update upgrades. Extensive experiments on large-scale intra-tissue, inter-tissue, and inter-data benchmarks show that scROD can achieve a better trade-off between model stability and plasticity than other state-of-the-art scRNA-seq annotation methods.

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

# A  APPENDIX

Table 5: The detailed information of all used datasets in our experiments.

| Data | Tissue | Technique | Cell type number | Cell number | Reference |
|------|--------|-----------|------------------|-------------|-----------|
| Baron_human | Pancreas | inDrop | 9 | 8569 | (Baron et al., 2016) |
| Enge | Pancreas | Smart-seq2 | 6 | 2282 | (Enge et al., 2017) |
| Muraro | Pancreas | CEL-Seq2 | 7 | 2122 | (Muraro et al., 2016) |
| Segerstolpe | Pancreas | Smart-seq2 | 6 | 1070 | (Segerstolpe et al., 2016) |
| Cao_Eye | Eye | sci-RNA-seq3 | 11 | 51836 | (Cao et al., 2020a) |
| Cao_Intestine | Intestine | sci-RNA-seq3 | 12 | 51650 | (Cao et al., 2020a) |
| Cao_Pancreas | Pancreas | sci-RNA-seq3 | 13 | 45653 | (Cao et al., 2020a) |
| Cao_Stomach | Stomach | sci-RNA-seq3 | 12 | 12106 | (Cao et al., 2020a) |
| He | Lone Bone | 10x Genomics | 11 | 15680 | (He et al., 2021) |
| Madissoon | Lung | 10x | 17 | 57020 | (Madissoon et al., 2020) |
| Stewart | Kidney | 10x | 18 | 26628 | (Stewart et al., 2019) |
| Vento | Placenta | 10x | 17 | 64734 | (Vento-Tormo et al., 2018) |

Table 6: Summary of five baseline methods for comparison.

| | Method | Year | Programming | Download URL | Reference |
|---|--------|------|-------------|--------------|-----------|
| Annotation | scNym | 2020 | Python | https://www.github.com/calico/scnym | (Kimmel & Kelley, 2021) |
| | scArches | 2022 | Python | https://github.com/theislab/scarches | (Lotfollahi et al., 2022) |
| | CIForm | 2023 | Python | https://github.com/zhanglab-wbgcas/CIForm | (Xu et al., 2023) |
| | scDOT | 2024 | Python | https://github.com/Zhangxf-ccnu/scDOT | (Xiong & Zhang, 2024) |
| Online integration | SCALEX | 2022 | Python | https://github.com/jsxlei/SCALEX | (Xiong et al., 2022) |

Table 7: Comparative analysis of performance among diverse baselines in intra-tissue continual compatible annotation benchmark.

| | Task 1 (Segerstolpe) | | | Task 2 (Muraro) | | | Task 3 (Enge) | | | Task 4 (Baron_human) | | |
|---|---|---|---|---|---|---|---|---|---|---|---|---|
| Pancreas tissue | old | new | overall | old | new | overall | old | new | overall | old | new | overall |
| **Finetune** | - | 93.3 | 93.3 | 92.7 | 96.1 | 95.0 | 90.5 | **96.7** | 93.1 | 94.9 | **98.2** | 96.9 |
| **Joint** | - | 93.3 | 93.3 | 95.6 | 96.9 | 96.5 | 95.9 | 96.5 | 96.2 | **96.8** | 97.6 | 97.3 |
| **scNym** | - | 93.3 | 93.3 | **99.0** | 96.6 | 97.4 | 95.1 | 94.9 | 95.0 | 97.1 | 96.8 | 96.9 |
| **scArches** | - | 62.1 | 62.1 | 84.3 | 84.1 | 84.2 | 74.7 | 78.7 | 76.4 | 93.1 | 93.3 | 93.2 |
| **SCALEX** | - | 74.6 | 74.6 | 83.8 | 84.8 | 84.5 | 78.0 | 84.4 | 80.8 | 92.5 | 93.2 | 92.9 |
| **CIForm** | - | 96.1 | 96.1 | 82.3 | 95.4 | 86.2 | 82.7 | 95.5 | 87.9 | 85.8 | 96.0 | 91.7 |
| **scDOT** | - | 80.4 | 80.4 | 75.9 | 90.3 | 82.6 | 71.3 | 88.1 | 78.9 | 83.2 | 94.5 | 88.6 |
| **Replay** | - | 93.3 | 93.3 | 95.9 | 96.9 | 96.6 | 96.4 | 96.8 | 96.6 | **96.8** | 97.8 | 97.4 |
| **scROD** | - | **98.6** | **98.6** | **99.0** | 97.0 | 97.6 | 97.7 | 96.7 | **97.3** | 96.7 | 98.0 | **97.5** |

## A.1  ADDITIONAL DETAILS

**Basic framework of scROD.** First, considering the discrete, sparse, and large variance characteristics of scRNA-seq data, we use the zero-inflated negative binomial (ZINB) distribution to model this gene

Table 8: Comparative analysis of performance among diverse baselines in inter-tissue continual compatible annotation benchmark.

| Cao atlas | Task 1 (Stomach) | | | Task 2 (Pancreas) | | | Task 3 (Intestine) | | | Task 4 (Eye) | | |
|---|---|---|---|---|---|---|---|---|---|---|---|---|
| | old | new | overall | old | new | overall | old | new | overall | old | new | overall |
| **Finetune** | - | 96.7 | 96.7 | 11.2 | **97.7** | 83.9 | 32.2 | **98.1** | 64.4 | 20.5 | **98.6** | 40.3 |
| **Joint** | - | 96.7 | 96.7 | **88.8** | 97.4 | **96.0** | **95.5** | 95.2 | **95.4** | **95.0** | 97.3 | **95.6** |
| **scNym** | - | 96.7 | 96.7 | 73.7 | 96.9 | 93.2 | 75.1 | 86.9 | 80.9 | 65.4 | 97.1 | 73.4 |
| **scArches** | - | 96.5 | 96.5 | 64.6 | 97.3 | 92.0 | 85.9 | 94.6 | 90.1 | 56.6 | 96.8 | 66.8 |
| **SCALEX** | - | 93.0 | 93.0 | 60.3 | 95.2 | 89.6 | 82.8 | 79.2 | 81.0 | 90.1 | 65.4 | 83.8 |
| **CIForm** | - | 97.2 | 97.2 | 70.4 | 96.1 | 92.3 | 79.8 | 90.3 | 84.7 | 61.5 | 97.6 | 70.3 |
| **scDOT** | - | 93.4 | 93.4 | 61.9 | 95.8 | 90.7 | 77.6 | 84.8 | 79.8 | 58.1 | 94.5 | 67.2 |
| **Replay** | - | 96.7 | 96.7 | 65.1 | **97.7** | 92.5 | 82.3 | 97.4 | 89.7 | 89.9 | 98.3 | 92.0 |
| **scROD** | - | **97.8** | **97.8** | 83.5 | **97.7** | 95.5 | 89.3 | 96.4 | 92.8 | 92.9 | 97.4 | 94.0 |

Table 9: Comparative analysis of performance among diverse baselines in inter-data continual compatible annotation benchmark.

| Mixed atlas | Task 1 (Vento) | | | Task 2 (Stewart) | | | Task 3 (Madissoon) | | | Task 4 (He) | | |
|---|---|---|---|---|---|---|---|---|---|---|---|---|
| | old | new | overall | old | new | overall | old | new | overall | old | new | overall |
| **Finetune** | - | 98.0 | 98.0 | 30.6 | **95.9** | 48.0 | 10.2 | **91.2** | 42.5 | 19.5 | **80.0** | 25.1 |
| **Joint** | - | 98.0 | 98.0 | **97.9** | 93.3 | **96.7** | **96.5** | 89.5 | **93.7** | **93.7** | 79.5 | **92.4** |
| **scNym** | - | 98.3 | 98.3 | 93.4 | 89.3 | 92.3 | 83.4 | 77.6 | 81.1 | 59.1 | 75.0 | 60.6 |
| **scArches** | - | 97.4 | 97.4 | 89.6 | 87.2 | 89.0 | 85.1 | 79.3 | 82.8 | 58.3 | 72.9 | 59.6 |
| **SCALEX** | - | 97.3 | 97.3 | 94.7 | 65.4 | 86.9 | 84.2 | 63.9 | 76.1 | 87.2 | 12.9 | 80.3 |
| **CIForm** | - | 97.7 | 97.7 | 91.2 | 92.9 | 92.6 | 84.5 | 78.4 | 82.0 | 63.7 | 74.2 | 65.8 |
| **scDOT** | - | 96.5 | 96.5 | 87.4 | 90.1 | 88.6 | 78.6 | 75.3 | 77.2 | 56.9 | 73.4 | 59.2 |
| **Replay** | - | 98.0 | 98.0 | 93.3 | 95.4 | 93.8 | 80.8 | 91.2 | 84.9 | 87.0 | 79.9 | 86.4 |
| **scROD** | - | **98.4** | **98.4** | 95.3 | 95.2 | 95.3 | 87.3 | 90.7 | 88.6 | 89.2 | **80.0** | 88.3 |

expression pattern, that is:

$$p_{zinb}(x_{ij}^*|\pi_{ij}, \mu_{ij}, \theta_{ij}) = \pi_{ij}\delta_{x_{ij}^*=0} + (1 - \pi_{ij}) \times \qquad (9)$$

$$\frac{\Gamma(x_{ij}^* + \theta_{ij})}{\Gamma(x_{ij}^* + 1)\Gamma(\theta_{ij})} \times (\frac{\theta_{ij}}{\theta_{ij} + \mu_{ij}})^{\theta_{ij}} \times (\frac{\mu_{ij}}{\theta_{ij} + \mu_{ij}})^{x_{ij}^*}.$$

Among them, $x_{ij}^*$ represents the raw read counts of the $i$-th cell on the $j$-th gene. $\pi_{ij}$, $\mu_{ij}$, $\theta_{ij}$ represent the zero-inflated parameters, mean parameters, and dispersion parameters, respectively, and they constitute the parameters to be estimated for the model.

Due to the complex interaction between genes, these three sets of parameters are not independent of each other but actually fall into a low-dimensional manifold. Therefore, we use the DCA model to estimate the parameters, and at the same time, to approximate the manifold, so as to effectively reduce the dimension and denoise the scRNA-seq data (Eraslan et al., 2019). Specifically, let $h_\theta(x) : R^m \to R^d$ be the encoder function that maps the cells into the low-dimensional embedding space and gets the embedding representation $z = h_\theta(x)$. Similarly, let $h_\theta^d(x) : R^d \to R^m$ be the decoder function and get the reconstructed variable $x_r = h_\theta^d(z)$. Then we use the reconstruct variable $x_r$ to estimate the parameters:

$$\hat{\pi} = sigmoid(w_\pi' x_r); \; \hat{\theta} = exp(w_\theta' x_r); \; \hat{\mu} = exp(w_\mu' x_r) \qquad (10)$$

where $w_\pi$, $w_\theta$, $w_\mu$ are the corresponding weights. Given the parameters, we can assume that the conditional distribution of the reconstructed data is independent, so we can use the negative log-likelihood of ZINB distribution as the first loss function:

$$\mathcal{L}_{zinb} = - \sum_{i=1}^{n_r+n_t} \sum_{j=1}^{m} p(x_{ij}^*|\hat{\pi}_{ij}, \hat{\mu}_{ij}, \hat{\theta}_{ij}). \qquad (11)$$

Table 10: Time-consuming analysis among diverse baselines on the large-scale inter-tissue benchmark.

| Cao atlas | Task 1 (Eye) | Task 2 (Intestine) | Task 3 (Pancreas) | Task 4 (Stomach) |
|---|---|---|---|---|
| **Finetune** | 454 | 876 | 983 | 1326 |
| **Joint** | 450 | 1248 | 2157 | 3496 |
| **scNym** | **312** | **598** | **726** | **1014** |
| **scArches** | 635 | 1204 | 1387 | 1859 |
| **SCALEX** | 706 | 1321 | 1569 | 2248 |
| **CIForm** | 386 | 752 | 961 | 1387 |
| **scDOT** | 581 | 1095 | 1302 | 1735 |
| **Replay** | 487 | 962 | 1095 | 1502 |
| **scROD** | 493 | 971 | 1108 | 1521 |

Table 11: Comparative analysis of performance among diverse baselines on MCL datasets.

| | pre-treatment | | | post-treatment | | |
|---|---|---|---|---|---|---|
| Method | old | new | overall | old | new | overall |
| **scNym** | - | 86.3 | 86.3 | 52.7 | 89.1 | 74.5 |
| **scArches** | - | 81.2 | 81.2 | 40.9 | 85.8 | 67.4 |
| **SCALEX** | - | 75.2 | 75.2 | 36.6 | 78.5 | 62.9 |
| **CIForm** | - | 84.1 | 84.1 | 48.3 | 87.4 | 71.7 |
| **scDOT** | - | 78.5 | 78.5 | 42.2 | 81.6 | 65.3 |
| **scROD** | - | **92.7** | **92.7** | **85.4** | **95.2** | **91.6** |

Actually, using data reconstruction as another kind of regularization can help reveal the global probabilistic structure of the whole dataset (Lopez et al., 2018; Chen et al., 2020).

In order to assign an annotation label for each cell, we attach a prototype-based classifier $f_\phi$ to the embedding layer. Take $t$-th period for example, $f_\phi$ projects the l2 normalized embedding $z_i$ into one of the $| \cup_{l=1}^t \mathcal{C}_l |$ cell types together with a similarity vector $s_r$, where $s_i = V z_i$ and $V = [v_1, v_2, ..., v_{|\cup_{l=1}^t \mathcal{C}_l|}]^T$ is the $l_2$ normalized prototype matrix. Then the annotation logits $o_i$ is obtained by regularizing $s_i$.

## A.2 DATA INFORMATION

The details of the twelve single-cell RNA sequencing (scRNA-seq) datasets employed in our investigations are comprehensively presented in Table 5. These experiments encompass intra-tissue analyses, as well as inter-tissue and inter-dataset comparisons. Each dataset features a cellular count exceeding 10,000 and encompasses a diversity of cell types, with a minimum of ten distinct types identified in any given set. Furthermore, these datasets originate from a range of organs and have been sequenced utilizing various platforms, highlighting the heterogeneity of the data sources in our study.

## A.3 DATA PREPROCESSING

For data preprocessing, we first normalize the total gene expression of each cell to 1e6, and then perform logarithmic transformation on the normalized data. Then we screen the top 2000 highly variable genes for training by default. Finally, we perform a z-score transformation for each gene in the training data. For the first training stage, there is no memory buffer at this time. We only need to select the highly variable genes of the training data in the first stage as model input. It is noted that our memory buffer stores the original gene expression and cell type labels of the cells. Starting from the second training stage, the single-cell data obtained in the current stage needs to be integrated with the single-cell data stored in the memory buffer. The principle of the integration is to select their intersecting genes as common features, and then perform data preprocessing and screening of highly variable genes based on these common features. Such a procedure has taken into account the state of data streams in real-world scenarios when they are continuously obtained.

Table 12: Comparative analysis of performance on the spatial data.

| | Tonsil | | | BE | | |
|---|---|---|---|---|---|---|
| Method | old | new | overall | old | new | overall |
| **STELLAR** | - | **92.5** | **92.5** | 81.2 | **90.4** | 84.7 |
| **scROD** | - | 92.4 | 92.4 | **88.6** | 90.1 | **89.0** |

Table 13: Comparative analysis of performance between different continual learning methods.

| | inter-tissue | | | inter-data | | |
|---|---|---|---|---|---|---|
| Choice | old | new | overall | old | new | overall |
| **(sc)SCR** | 86.4 | **92.9** | 87.0 | 74.8 | 96.6 | 84.5 |
| **(sc)ACE** | 87.2 | 92.5 | 87.6 | 76.1 | 95.8 | 84.9 |
| **scROD** | **89.3** | 92.6 | **89.5** | **78.8** | **97.0** | **86.1** |

## A.4 Additional results

**Intra-tissue annotation** Given the constraints of space, we have included only a single instance of the tissue-based data stream within the main body of the text. The outcomes from an alternate sequential scenario are presented in Table 7. Remarkably, this scenario depicts a sequence that is entirely inverse to the one discussed in the text. Even when the sequence of data learning is inverted, scROD consistently outperforms other benchmark methods, including Joint, by significant margins, which further underscores its superiority as detailed in the text. This highlights scROD's robustness in the context of continual learning compatibility. The superior performance of scROD compared to all benchmarks during the initial phase of learning underscores its exceptional proficiency in performing foundational annotations. Its sustained performance in subsequent phases underscores the strategy's efficacy in mitigating catastrophic forgetting through mechanisms such as sample replay and objective decomposition. A closer comparison between scROD and Joint reveals that although Joint retains all training samples, it fails to offer competitive results. This discrepancy suggests that objective decomposition may play a more pivotal role in preventing forgetting than merely retaining a larger sample size.

To gain a clearer visualization of scROD's learning progress following task completion, we extracted its low-dimensional embedding features. Subsequently, we applied the Uniform Manifold Approximation and Projection (UMAP) methodology to visually represent these features within a two-dimensional space, thus facilitating an intuitive understanding of the learning situation. Figure 6 shows the UMAP plots of scROD after each learning task. The findings indicate that scROD successfully retains the knowledge of previously learned cell types while concurrently acquiring new tasks, thus showcasing its exceptional performance in continual compatible learning. This is particularly evident in its capacity to accurately classify both historical and recently introduced cell types. Notably, scROD demonstrates robust recognition and retention abilities even for cell types represented by smaller sample sizes, including PP cells, macrophages, and endothelial cells. These results underscore scROD's capability to strike an effective balance between learning new information (plasticity) and preserving existing knowledge (stability), reinforcing its potential as a tool for advancing the field of continual learning in single-cell type classification.

**Inter-tissue annotation.** Similarly, we have included only a single instance of the tissue-based data stream in the main body of the text. The outcomes from an additional sequential scenario are illustrated in Table 8. Relative to its efficacy in intra-tissue experiments, the performance of scROD in inter-tissue assays has demonstrated a degree of consistency across the three measures of accuracy, with no substantial decline. This finding underscores the robustness of scROD in mitigating batch effects. Conversely, the alternate baseline models demonstrated a marked reduction in accuracy, notably in terms of retaining previously acquired information. This decline in 'old accuracy' could be attributed to the amplification of batch effects arising from the heterogeneous data assimilated during distinct task learning phases, thereby exacerbating the model's challenge in preserving its acquired knowledge. Our observations indicate that the Joint approach consistently outperforms in terms of both old and overall accuracy, a result attributed to its comprehensive caching of samples. Nevertheless, the practicality of this strategy is constrained by limited available memory, rendering

Table 14: Performance comparison among diverse baselines on the CIFAR-10 and CIFAR-100 datasets.

| | CIFAR-10 | | CIFAR-100 | |
| ResNet18 | buffer=500 | buffer=5000 | buffer=500 | buffer=5000 |
|---|---|---|---|---|
| Finetune | 19.7 | 19.7 | 14.7 | 14.7 |
| Joint | **91.8** | **91.8** | **70.1** | **70.1** |
| Replay | 61.7 | 83.6 | 27.7 | 53.9 |
| (sc)ROD | 68.8 | 85.9 | 41.5 | 58.6 |

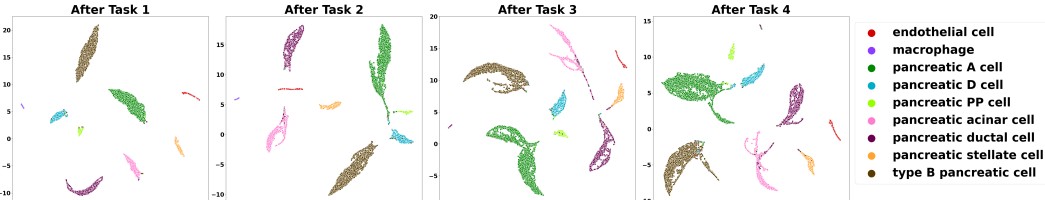

Figure 6: The UMAP plots of scROD after each task on the intra-tissue benchmark. Baron_human, Enge, Muraro, and Segerstolpe are serialized as four sequential learning tasks.

it less feasible for extensive application. Moreover, the predilection of the Joint method to retain an excessive number of samples from prior tasks has a discernible impact on its capacity to acquire new information. Consequently, when evaluated on its proficiency in learning novel tasks, Joint exhibits a marginal underperformance compared to the scROD method in terms of new accuracy metrics. scROD exhibited superior performance, outperformed only by Joint, in terms of both old and overall accuracy, underscoring the success of its approach in mitigating catastrophic forgetting and eliminating batch differences. The outstanding achievement of scROD in maintaining high levels of accuracy for both previously encountered and novel datasets emphasizes the significance of its objective decomposition strategy as a means of achieving an optimal balance between stability and adaptability. Thus, although the order of data learning is reversed, scROD still has excellent performance in the inter-tissue experiment.

**Inter-data annotation.** In the context of this benchmark, we have identified and chosen four extensive datasets—namely He, Madissoon, Stewart, and Vento—each of which has been sequenced utilizing distinct tissues and technologies. This diverse selection will enable a comprehensive evaluation across a variety of sequencing parameters and biological samples. The manuscript presents one representative instance from these four datasets within the main text, while the corresponding experimental outcomes for an additional case are delineated in Table 9. The presence of a batch effect introduces a level of intimidation across the performance of all methodologies employed. Despite this challenge, scROD retains a commendable level of performance, illustrating its proficiency in integrating cells from diverse datasets into a coherent embedding space. Additionally, the observed consistency in accuracy suggests that scROD's capacity to mitigate catastrophic forgetting remains unimpaired by batch effects. This finding underscores the method's resilience and adaptability within varying experimental conditions. The observed data reveal a noteworthy trend where, with the exception of baseline Joint, alternative baselines exhibit a diminished level of competitiveness, particularly with respect to old accuracy metrics. This decline in performance becomes increasingly pronounced in correlation with the augmentation of the number of datasets subjected to the learning process. While the Joint approach yields impressive results in inter-dataset experiments, its methodology, which involves retaining all previously learned samples, is not recommended due to potential scalability and efficiency issues. In contrast, scROD consistently exhibits superior performance in the context of continuous learning that is compatible with dynamic data streams. This advantage has led to its widespread adoption in practical applications of single-cell annotation.

**Statistical Analysis.** In order to prove the consistency and stability of the results of our method, we report their standard deviation values. Corresponding to Table 1, Table 2 and Table 3 in the text, the standard deviations of three runs results are within the interval (0.3, 1.1) for scROD, which fluctuates relatively little. We also conduct the significance test of the improvements in results. Specifically, we choose the first two best-performing baselines Joint and scNym to perform the one-sided pairwise t-test with scROD on the overall accuracy. The p-values are 0.910 (scROD vs Joint) and 0.002

(scROD vs scNym), demonstrating that the improvement of scROD compared to scNym is significant, and the performance of scROD and Joint is comparable.

**Time-consuming analysis.** Here we give the average running time of each method on the large-scale inter-tissue benchmark in Table 10. It can be seen that scROD and Replay methods hold almost the same magnitude of running costs as the Finetune strategy, much lower than the Joint strategy. In addition, as the number of tasks increases, the time consumed by the Joint method is twice that consumed by the Finetune, Replay, and scROD. Although scNym and CIFOrm consume the smallest computational cost, their performance cannot be competitive with our method. In general, the combination of efficiency and performance shows the advantages of our approach to solving this task.

**Application in longitudinal data.** Here We apply scROD to a multi-timepoint longitudinal single-cell dataset, i.e., mantle cell lymphoma (MCL) dataset (Zhang et al., 2021). Since the timing of measurements varies from patient to patient, we manually binarize the time variable into two groups: pre-treatment and post-treatment, which also aligns with the analysis in the original paper. We first train each method in the pre-treatment group and then continually train models in the post-treatment group. The labeled ratio is set to 0.1 by default. The results in Table 11 show that our method can consistently outperform other baselines in the longitudinal data situation.

**Application in spatial data.** Our method can be extended to spatial data by simply replacing the model backbone with a network that adapts to spatial data, such as the graph neural network. Here we select two single-cell spatial data, i.e., Tonsil and BE datasets (Goltsev et al., 2018). Then we use the same data preprocessing and model backbone as in STELLAR (Brbić et al., 2022). We first train the model on the Tonsil dataset and continually train the model on the BE dataset. The labeled ratio is also set to 0.1 by default. The results in Table 12 show that once we enter the second stage, STELLAR will lose some accuracy on the Tonsil dataset, but our method can alleviate this problem and achieve higher accuracy on the two datasets.

**Comparison with continual algorithms.** Here we select two representative continual learning algorithms in the machine learning community, i.e., supervised contrastive replay (SCR) (Mai et al., 2021) and asymmetric cross-entropy (ACE) (Caccia et al., 2022). They also use the memory buffer and have customized designs in training loss functions for continual learning tasks. We run these algorithms on inter-tissue and inter-data annotation benchmarks. The results after the fourth training stage are shown in the Table 13. We can see that our loss decomposition strategy performs better than the other two continual learning methods in the trade-off between model stability and plasticity. It is reasonable because they mix the learning objectives of new/old cell type distinction and new cell type distinction.

**Application in other domain.** Since this paper aims to solve the problem of continual compatible annotation of scRNA-seq data, all experiments are focused on this data type for verification. In terms of the overall idea, our method is a general machine-learning approach that can be applied to continual learning tasks on different data types. To validate this claim, we choose two image classification datasets, i.e., CIFAR-10 and CIFAR-100, in the vision field for experiments. Following the task-setting in this field, two datasets consist of 5 disjoint tasks with each task having 2 and 20 classes, respectively. We report the average accuracy of all tasks after the last training stage. The results in Table 14 show that our method can be applied to the continual learning task in the vision field.

**Method limitation.** One limitation of scROD is that we need to maintain a lightweight memory buffer to replay a few samples during the compatible continual annotation process. Once these samples become unreachable due to data privacy, the memory buffer cannot be constructed. So our future work is to develop replay-free algorithms that eliminate the necessity for memory buffers.

