# OpenReview forum: "Compatibility-aware Single-cell Continual Annotation"
_ICLR.cc/2025/Conference — ICLR 2025 Conference Withdrawn Submission_

### Official Review · Reviewer_E1SE · 2024-10-31

**Soundness:** 2
**Presentation:** 2
**Contribution:** 1
**Rating:** 3
**Confidence:** 4

**Summary:**

The paper presents scROD, a method designed to address the challenge of updating automatic cell type annotation models in single-cell RNA-seq (scRNA-seq) data while preventing catastrophic forgetting, where the model's performance on previously learned tasks deteriorates after learning new tasks. To tackle this, the authors introduce the concept of continual compatible learning, which emphasizes maintaining stability on old tasks while adapting to new ones. The proposed scROD method leverages sample replay by using a memory buffer to retain cells from earlier tasks, allowing the model to learn these alongside new tasks. It also separates two training objectives: distinguishing new cell types from old ones and differentiating between newly introduced cell types. By assigning distinct weights to these objectives, scROD achieves a balance between stability and adaptability.

**Strengths:**

1. A thorough investigation of the continual cell type annotation task.
2. Provide comprehensive experimental benchmarks for the proposed method and baselines.

**Weaknesses:**

1. This manuscript focuses on one task: continual cell-type annotation. My main concern is that the importance of such continual annotation might not be very high. There exists some large-scale atlas and databases covering different species, like Human Cell Atlas, CELLxGENE, Mouse Cell Atlas, Zebrahub, and so on. Those resources cover a large range of tissues and provide cell type annotations. Some of them also curate the annotations with Cell Ontology. In most cases, a simple model pretrained on some atlas, such as CellTypist[1], can handle the annotation of unseen data. Can the authors provide concrete examples of scenarios where continual learning would be necessary or advantageous? What are the limitations of current approaches that continual learning specifically addresses?
2. Related to bullet 1, currently all the experiments only focus on continual cell type annotation. Have the authors considered evaluating their method on vanilla annotation or zero-/few-shot annotation tasks? How might the proposed method's performance compare to existing methods in these scenarios?
3. According to the experimental results, the performance of the proposed method is not always better than the baselines. If so, what factors contribute to these performance differences? How do the computational requirements of scROD compare to the baselines?

[1] Domínguez Conde, C., et al. "Cross-tissue immune cell analysis reveals tissue-specific features in humans." *Science* 376.6594 (2022): eabl5197.

**Questions:**

See weaknesses.

---

### Official Review · Reviewer_JBpo · 2024-11-04

**Soundness:** 3
**Presentation:** 3
**Contribution:** 2
**Rating:** 3
**Confidence:** 4

**Summary:**

Authors propose an online learning approach scROD to annotate single cell RNA seq data. scROD uses a memory buffer and a new loss function to preserve classification performance on the past data while being able to annotate newly acquired data at the same time. Authors compare with several baselines and existing methods demonstrating the improvements in continual single-cell annotation.

**Strengths:**

1. The paper is clearly written and the experiments are clearly presented to back the claims made by the authors.
2. The problem is interesting since many single-cell RNA datasets have come up in the past few years. Online learning or transfer learning which ensures that the same network/fine-tuned networks can successfully annotate new data would be a strong contribution to the scientific community.

**Weaknesses:**

1. The idea to utilize a memory buffer is a widely-used idea in the reinforcement learning literature (Deep Q-Networks) and even continual learning literature (Gradient Episodic Memory). Therefore the core contribution is not technically novel. There is some novelty to decompose the loss function and consider the impact of different loss functions on the catastrophic forgetting issue in this setting but the results are fairly obvious. For example, when we are training on new datasets, we should ensure class balancing to ensure no classes are compromised which can be achieved with weighing loss function or sampling per class.
2. There are no past methods that specifically target the problem of continual learning but rather consider query and reference datasets, which I believe is a much harder problem with no supervision available on a query dataset. Therefore comparison with these methods is good to have but unfair to evaluate the utility of scROD.

**Questions:**

1. How is scROD different from GEM (Gradient Episodic Memory, Lopez-Paz, D., & Ranzato, M. A., 2017)? Can authors repurpose GEM and compare it with scROD? There are several follow ups for GEM for example "Adaptive Memory Replay for Continual Learning" from James et. al 2024 and "MGSER-SAM: Memory-Guided Soft Experience Replay with Sharpness-Aware Optimization for Enhanced Continual Learning" from Li et. al 2024 that could be compared against? Can authors theoretically and experimentally compare scROD with these approaches?
2. Since the manuscript tackles annotating scRNA-seq datasets, are there any practical limitations of this approach, can this be directly deployed by medical practitioners? There already exist many methods for annotation which do no assume access to supervision on query datasets, how did the authors consider online setting relevant to scRNA-seq dataset annotation? Is it possible to get small labeled samples on a query dataset?
3. Can you make accurate biological inferences from this method? Is it possible to identify genes which cause classification to a particular cell-type?

---

### Official Review · Reviewer_RNHg · 2024-11-05

**Soundness:** 2
**Presentation:** 3
**Contribution:** 2
**Rating:** 5
**Confidence:** 4

**Summary:**

The paper introduces scROD, a method for continual compatible learning in the context of single-cell RNA sequencing (scRNA-seq) data annotation. scROD employs a combination of sample replay and objective decomposition to address the challenge of catastrophic forgetting, where models typically lose performance on old tasks after learning new ones. By maintaining a memory buffer to store samples from previous tasks and replaying them alongside new data, scROD balances the retention of old knowledge with the acquisition of new information. Furthermore, it decomposes the training objectives into new/old cell type distinction and new cell type distinction, assigning different weights to these objectives to achieve a better trade-off between model stability and plasticity. This approach allows scROD to continuously learn and annotate new cell types over time without forgetting previously learned ones, demonstrating effectiveness through comprehensive experiments on various benchmarks.

**Strengths:**

1. The paper presents a novel framework dubbed scROD that combines sample replay and objective decomposition, addressing the critical issue of catastrophic forgetting in continual learning scenarios. scROD effectively balances the model's ability to retain old knowledge (stability) and adapt to new tasks (plasticity), which is crucial for continual learning systems.

2. The paper evaluates scROD on a variety of benchmarks, including intra-tissue, inter-tissue, and inter-data scenarios, demonstrating its robustness across different annotation challenges. Besides, scROD outperforms existing state-of-the-art methods in scRNA-seq annotation, showing significant improvements in both old and new task accuracies.

3. The article presents its findings with clear and concise figures and tables.

**Weaknesses:**

1. The innovation of this paper is quite common in continual learning, where many methods use replay buffer approaches to tackle catastrophic forgetting (e.g., [R1][R2][R3]). This paper does not show significant differences from those methods or specific distinctions for RNA data.

2. The novelty of objective decomposition is intuitive and easy to understand, which leverage two parameters \alpha_1 and \alpha_2 to balance the optimization objectives.

3. The experimental analysis of two learning objectives is trivial, since derivation of Eq. 7 is intuitive. Besides, it seems like that only using L_cur leads to better performance on previous tasks than L_pre, as shown in Figure 3. Thus, why do not just leveraging L_cur instead of L_pre?

4. Although ScROD achieves SOTA performance in various settings, Tables 1, 2 and 3 show that ScROD achieves only a little bit higher than Replay, which is not the compelling evidence of the effectiveness of ScROD.

5. In Figure 5, the first two experiments were performed on inter-data benchmark , and the last two on inter-tissue benchmark. Why not using the same benchmark for all the ablation study?


[R1] Maracani, A., Michieli, U., Toldo, M., & Zanuttigh, P. (2021). Recall: Replay-based continual learning in semantic segmentation. In Proceedings of the IEEE/CVF international conference on computer vision (pp. 7026-7035).

[R2] Chaudhry, A., Rohrbach, M., Elhoseiny, M., Ajanthan, T., Dokania, P. K., Torr, P. H., & Ranzato, M. A. (2019). On tiny episodic memories in continual learning. arXiv preprint arXiv:1902.10486.

[R3] Riemer, M., Cases, I., Ajemian, R., Liu, M., Rish, I., Tu, Y., & Tesauro, G. (2018). Learning to learn without forgetting by maximizing transfer and minimizing interference. arXiv preprint arXiv:1810.11910.

**Questions:**

Please see the Weaknesses section.

---

### Note · Authors · 2024-11-12

I have read and agree with the venue's withdrawal policy on behalf of myself and my co-authors.